# Impact of Interrepetition Rest on Muscle Blood Flow and Exercise Tolerance during Resistance Exercise

**DOI:** 10.3390/medicina58060822

**Published:** 2022-06-18

**Authors:** Jayson Gifford, Jason Kofoed, Olivia Leach, Taysom Wallace, Abigail Dorff, Brady E. Hanson, Meagan Proffit, Garrett Griffin, Jessica Collins

**Affiliations:** 1Department of Exercise Sciences, Brigham Young University, Provo, UT 84602, USA; jason.s.kofoed@gmail.com (J.K.); olivia.k.leach@gmail.com (O.L.); taysomew@gmail.com (T.W.); adorff.16@gmail.com (A.D.); brady.hanson22@gmail.com (B.E.H.); meagan.proffit@gmail.com (M.P.); garrgriff.2015@gmail.com (G.G.); young.jess12@gmail.com (J.C.); 2Program of Gerontology, Brigham Young University, Provo, UT 84602, USA

**Keywords:** reactive hyperemia, cluster set, strength training, muscle oxygenation

## Abstract

***Background and Objectives:*** Muscle blood flow is impeded during resistance exercise contractions, but immediately increases during recovery. The purpose of this study was to determine the impact of brief bouts of rest (2 s) between repetitions of resistance exercise on muscle blood flow and exercise tolerance. ***Materials and Methods:*** Ten healthy young adults performed single-leg knee extension resistance exercises with no rest between repetitions (i.e., continuous) and with 2 s of rest between each repetition (i.e., intermittent). Exercise tolerance was measured as the maximal power that could be sustained for 3 min (P_SUS_) and as the maximum number of repetitions (Reps_80%_) that could be performed at 80% one-repetition maximum (1RM). The leg blood flow, muscle oxygenation of the vastus lateralis and mean arterial pressure (MAP) were measured during various exercise trials. Alpha was set to *p* ≤ 0.05. ***Results:*** Leg blood flow was significantly greater, while vascular resistance and MAP were significantly less during intermittent compared with continuous resistance exercise at the same power outputs (*p* < 0.01). P_SUS_ was significantly greater during intermittent than continuous resistance exercise (29.5 ± 2.1 vs. 21.7 ± 1.2 W, *p* = 0.01). Reps_80%_ was also significantly greater during intermittent compared with continuous resistance exercise (26.5 ± 5.3 vs. 16.8 ± 2.1 repetitions, respectively; *p* = 0.02), potentially due to increased leg blood flow and muscle oxygen saturation during intermittent resistance exercise (*p* < 0.05). ***Conclusions:*** In conclusion, a brief rest between repetitions of resistance exercise effectively decreased vascular resistance, increased blood flow to the exercising muscle, and increased exercise tolerance to resistance exercise.

## 1. Introduction

Previous studies demonstrated that providing brief bouts of rest between repetitions (i.e., interrepetition rest or cluster sets) of resistance exercises (e.g., high intensities of bench press or back squats) delays the development of fatigue and power loss [1,2,3,4] compared with traditional configurations with no rest between repetitions. In addition to greater fatigue resistance, studies reported that interrepetition rest also reduces the cardiovascular burden of resistance exercise with lower increases in blood pressure being observed when a brief rest is allocated between repetitions of high-intensity back squats and leg extensions [5,6]. It seems possible that improvements in fatigue development and cardiovascular strain associated with interrepetition rest may be related to enhanced muscle blood flow.

In addition to being essential for aerobic ATP resynthesis, the delivery of blood to an active muscle facilitates the resynthesis of creatine phosphate (CrP) and clearing metabolites from the muscle [7], both of which are very important during resistance exercise. In fact, when the delivery of fresh blood to an active muscle is prevented, CrP resynthesis does not occur [7] and exercise tolerance is dramatically reduced [8,9].

During resistance exercise, intramuscular pressure reaches levels that are sufficient to impinge upon the small vessels within the muscle [10,11]. Consequently, blood flow is impeded [10,12,13] and vascular resistance and blood pressure reach very high levels during resistance-type contractions [11,14,15]. This reduction in muscle blood flow likely contributes to the decrease in sustainability (i.e., the number of repetitions that can be performed) that occurs as contraction intensity increases [16,17]. 

While the impediment of blood flow to the active muscle lasts as long as the contraction is performed [10], blood flow to the active muscle is restored when the contraction ceases. Following the release of a muscle contraction, a process called reactive hyperemia occurs [18] in which previously impinged blood vessels are reopened and a backlog of blood flow rapidly perfuses the downstream muscle, increasing flow by more than 200% above resting values within a single heartbeat and by more than 400% within five heartbeats [10,19].

Considering the rapid increase in blood flow that occurs following the release of a single contraction, reactive hyperemia likely occurs during intermittent rest. While several lines of evidence suggest that the improved exercise tolerance and cardiovascular strain associated with interrepetition rest are likely related to intermittent decreases in vascular resistance and improved blood flow to the exercising muscle [1,3,6,20], this hypothesis has not been directly tested. Therefore, the purpose of this study was to determine the impact of brief bouts of rest between repetitions of resistance exercise on muscle blood flow and exercise tolerance. We hypothesized that 2 s bouts of rest between repetitions of knee extension exercise would be sufficient to decrease vascular resistance, increase muscle blood flow and improve exercise tolerance compared with traditional continuous resistance exercise.

## 2. Methods

### 2.1. General Overview of Experimental Approach

Two separate study protocols of single-leg knee extension exercise, each with two configurations of repetitions (continuous exercise with no rest between repetitions and intermittent exercise with 2 s of interrepetition rest), were used to determine the impact of interrepetition rest on exercise tolerance and hemodynamics during various intensities of single-leg knee extension exercise. The major independent variables were the exercise configuration (i.e., continuous exercise or intermittent exercise with interrepetition rest) and power output of exercise. The major dependent variables included the total repetitions that could be performed, total work performed, leg blood flow, mean arterial pressure, vascular resistance and muscle oxygenation. Major comparisons were made within subjects on data collected on the same day (i.e., repeated measures comparing continuous vs. intermittent exercise performed on the same day). Each study protocol was performed on separate days with at least 48 h of rest between protocols. As described in greater detail below, the first protocol focused on the impact of interrepetition rest on exercise tolerance and hemodynamics during lighter-weight, higher-repetition resistance exercise, while the second protocol focused on the impact of interrepetition rest on exercise tolerance and hemodynamics during heavier-weight resistance exercise.

### 2.2. Subjects

This study was approved by the Institutional Review Board at Brigham Young University. All subjects provided written informed consent prior to participation in this study. This study was not registered on a clinical trials database.

Ten young (age 23.0 ± 0.6 years), healthy adults (4 female, 6 male) with an average height of 1.73 ± 0.36 m, body mass of 78.84 ± 5.57 kg, body mass of index of 25.78 ± 1.2 kg∙m^−2^ and an average knee extension 1RM of 66.52 ± 5.45 kg participated in this study. All subjects were healthy, non-smokers and free from cardiovascular disease. Subjects from a variety of self-reported training backgrounds (e.g., sedentary, endurance training and strength training) were used for this study.

### 2.3. Procedures

All testing was performed in the Human Performance Research Laboratory at Brigham Young University. Exercise trials were supervised by research personnel trained to monitor the knee extension exercises described below, while the collection of physiological data was performed by research personnel experienced in making cardiovascular measurements during exercise.

Following a familiarization visit, subjects participated in 2 different protocols. Protocol #1 determined the peak power that could be sustained for 3 min (i.e., P_SUS_) during resistance exercise with and without interrepetition rest, while hemodynamics were measured. Protocol #2 measured the hemodynamic response and exercise tolerance to heavy weight (80% 1RM) resistance exercise with and without interrepetition rest.

Subjects reported to the lab well rested, having fasted for 4 h and having refrained from exercise and consumption of alcohol or caffeine for ~24 h before each visit [21]. With primary comparisons being made within subjects from data collected on the same day (i.e., within day comparisons), women completed their visits at any point during the menstrual cycle and not just during a narrow window of the cycle [22]. All data collection and exercise were completed on the subject’s right leg, regardless of leg dominance. Each protocol is described in greater detail below.

### 2.4. Familiarization Visit

Familiarization with the knee extension exercise: As illustrated in Figure 1, subjects were seated in a chair and had their leg attached to a pulley-based weight stack (NK664-75 DeLuxe wall pulley; NK Products, Lake Elsinore, CA, USA) that was situated to perform single-leg knee extension. This weight stack involved a cord that passed from the ankle of the subject through a series of pulleys to vertically displace a weight stack resting behind the seated subject. To standardize the movement within protocols the knee extension contraction started with the leg resting at a 90° flexed position and required 15 cm of leg extension before returning to the flexed position. Both concentric and eccentric actions were performed during the contraction. The leg was rested at 90° of knee flexion to minimize the impact that differing degrees of knee flexion have on the blood flow response to knee extension exercise [23]. Since the hyperemic response to knee extension is influenced by the amount of external work performed during a contraction [24,25], the displacement distance of the weight was limited to 15 cm for each contraction. Limiting the displacement distance to 15 cm also prevented the subjects from fully extending and “locking out” their knees, which could introduce undesired bouts of rest between concentric and eccentric contractions.

Determination of the knee extension one-repetition maximum (1RM): Following a brief familiarization with the exercise, the assessment of the knee extension 1RM was completed using the protocol prescribed by the National Strength and Conditioning Association (Sheppard & Triplett, 2016). Subjects performed a warmup of approximately 5–10 repetitions with a moderate weight, followed by ~3 min of rest and increasingly heavier lifts estimated to yield 2–3 repetitions. The weight was then increased an additional 10–20% and subjects were asked to complete one repetition. If the subjects successfully moved the weight through the full 15 cm of motion, the weight was increased by an additional 10% until task failure. The greatest weight lifted through a complete range of motion was defined as 1RM. Two to three minutes of rest separated each attempt.

### 2.5. Protocol #1: Determining the Effect of Interrepetition Rest on Muscle Blood Flow and Maximum Sustainable Power

Having previously been familiarized with the single-leg knee extension resistance exercise (described above), the subjects reported to the lab to perform a graded exercise test to determine the highest power output that could be sustained for 3 min (P_SUS_) of continuous (no interrepetition rest) and 3 min of intermittent (2 s of interrepetition rest) resistance exercise. The leg started in the same flexed position (90° knee flexion) with no tension from the weight on the leg for the continuous and intermittent protocols. The subsequent contraction phases of both protocols were identical, with subjects first performing a concentric knee extension to raise the weight stack a fixed distance of 15 cm in one second followed immediately by active eccentric knee flexion to lower the weight stack 15 cm to the original starting position in another one second. When the leg returned to the starting position (90° knee flexion) during continuous resistance exercise a subsequent contraction was immediately performed. In contrast, when the leg returned to the starting position during intermittent resistance exercise, 2 s of rest were given (leg still in the 90° flexed position with no tension from the weight) before performing the next contraction cycle. An audio recording prompted the subjects to contract and rest at the appropriate times.

The graded exercise test started with the subjects performing 3 min of continuous resistance exercise at 15 lbs. If a continuous bout of exercise was successfully completed, the subjects performed another bout of continuous exercise with an additional 7.5 lbs of weight after a 5 min rest period. This pattern continued until task failure. The subjects also performed an intermittent graded exercise test on the same visit. Following at least 5 min of rest, the subjects performed 3 min of intermittent resistance exercise at 15 lbs. If an intermittent bout of resistance exercise was successfully completed, another bout with an additional 15 lbs of weight was performed after a 5 min rest period. This pattern continued until task failure. The highest power output sustained for 3 min was identified as P_SUS_. With pilot studies indicating that task failure would occur at a lighter weight during continuous exercise than intermittent exercise, the continuous bouts of exercise were always performed before the intermittent bouts, thereby biasing the potential impact of fatigue induced by previous bouts toward the intermittent exercise (i.e., more work was performed prior to an intermittent bout of exercise compared with a bout of continuous exercise of equal power). 

As described in greater detail below, leg blood flow (Doppler ultrasound of the femoral artery) and mean arterial pressure (finger photoplethysmography) were measured during 1 min of rest in the starting position before any exercise had occurred and during the final 30 s of exercise for each bout. Comparisons of the hemodynamics were only made for power outputs that all subjects completed (i.e., 10 and 15 W).

### 2.6. Protocol #2: Determining the Impact of Interrepetition Rest on Exercise Tolerance and Blood Flow during Resistance Exercise with Heavy Weight

Subjects reported to the lab well rested (at least 48 h before or after protocol #1) to perform 3 bouts of heavy-weight resistance exercise to task failure. Following a brief warm-up, subjects performed bouts of continuous resistance exercise at 40% and 80% 1RM (c40% and c80%, respectively) and a bout of intermittent resistance exercise at 80% 1RM (i80%) until task failure. Thirty minutes of recovery separated the bouts, which were performed in random order. As was the case for protocol #1, task failure was defined as the inability to lift the weight stack the full 15 cm on two consecutive contractions or the inability to maintain the proper rate of contractions despite verbal encouragement.

The leg blood flow (Doppler Ultrasound), mean arterial pressure (finger photoplethysmography) and the oxygenation of the vastus lateralis (near-infrared spectroscopy (NIRS)) were measured during 1 min of rest prior to each bout and during the entirety of each exercise bout.

Measurement of the leg blood flow: Measurements of common femoral artery blood velocity and artery diameter were taken 2–3 cm proximal to the superficial/deep bifurcation using a Logiq E ultrasound Doppler system in duplex mode (General Electric Medical Systems, Milwaukee, WI, USA) equipped with a linear array transducer functioning at a B-mode frequency of 9 MHz and a Doppler frequency of 5 MHz. Blood velocity was assessed with an insonation angle of 60°, while the sample size was maximized and centered according to vessel size and position in real time. Femoral artery blood flow (ml·min^−1^) was calculated using the following equation [21]:$$\mathrm{Femoral}\;\mathrm{blood}\;\mathrm{flow}=\left[\left(\mathrm{mean}\;\mathrm{blood}\;\mathrm{velocity}\right)\times (\pi \times ({\mathrm{vessel}\;\mathrm{radius}}^{2}\right))\times 60\;s]$$
where mean blood velocity is expressed in cm·s^−1^ and radius is expressed in cm.

Measurement of the mean arterial pressure: Blood pressure was measured continuously with a finger photoplethysmography system (CNAP, CNSystems, Graz, Austria). MAP was calculated as the pressure–time integral of the continuous finger blood pressure measurement [21]. Data from the photoplethysmographer were relayed to a data acquisition program (Acknowledge, Biopac Systems Inc., Goleta, CA, USA) in real time. Vascular resistance was subsequently calculated as the quotient of MAP divided by the simultaneously measured leg blood flow [18].

Measurement of the muscle oxygenation: As described by Broxterman et al. [26], near-infrared spectroscopy (NIRS) was used to measure the concentration of muscle heme (i.e., [Heme]) and percent saturation of muscle heme during protocol #2. Specifically, the NIRS probe (Oxiplex TS, ISS instruments, Champaign, Il, USA) with light-detector separations distances of 2.0–3.5 cm was adhered to the skin directly superficial to the vastus lateralis (identified using ultrasound imaging) at mid-thigh. Flexible tape was subsequently wrapped around the leg and sensor to hold the sensor in place and to block external sources of light from reaching the detector. The probe remained adhered to the skin in the same position on the thigh throughout the entirety of the visit. The NIRS probe was calibrated using a block with known absorption and scattering coefficients. The calibration was subsequently confirmed on a separate block with different absorption and scattering coefficients. Data from the NIRS were relayed to a data acquisition program (Biopac) in real time. The subcutaneous adipose tissue thickness (measured by B-mode ultrasound) was less than 1.75 cm for all subjects.

Calculation of the work and power: Work per contraction cycle of resistance exercise, expressed as Joules per contraction cycle, was calculated as the product of the mass (kg) lifted multiplied by gravity (9.8 m∙s^−2^) and the total displacement distance (m) [27]. Displacement distance was calculated as the absolute value of the distance traveled by the weight stack during the concentric and eccentric phases of contraction (15 cm each way). Power, expressed as W, was subsequently calculated as the quotient of work per contraction divided by the duration of each contraction cycle, which was 2 s for continuous and 4 s for intermittent resistance exercise [27].

Statistical analysis: The Shapiro–Wilk test was used to determine whether the data of interest exhibited normality (*p* > 0.05). Paired t-tests were used to analyze differences in P_SUS_ between the continuous and intermittent resistance exercises in protocol #1. Repeated-measures ANOVA was used to identify differences in hemodynamics and exercise tolerance in protocols #1 and #2. In the event of a significant omnibus, paired *t*-tests of planned comparisons were used to identify which comparisons of interest were significantly different. Alpha was set at *p* ≤ 0.05 a priori. All statistical analyses were completed using SPSS version 26 (SPSS Inc., Chicago, IL, USA). Data are expressed as the mean ± SE.

## 3. Results

### 3.1. Protocol #1: The Effect of Interrepetition Rest during High-Repetition, Low-Weight Resistance Exercise

Hemodynamics during continuous vs. intermittent resistance exercise: The leg blood flow and mean arterial pressure were measured during the graded exercise tests. While subjects reached a range of P_SUS_ (see below), all subjects were able to complete the 10- and 15-W bouts with both continuous and intermittent configurations. Consequently, the hemodynamic response to 10 and 15 W during the continuous and intermittent configurations were compared.

As illustrated in Figure 2A, when considering blood flow during the two configurations of resistance exercise, repeated-measures ANOVA revealed significant main effects of power (F = 26.73.80, *p* < 0.01, partial $\eta^{2}=0.73$) and configuration (F = 7.95, *p* < 0.01, partial $\eta^{2}=0.44$) such that, on average, leg blood flow was lower during continuous versus intermittent resistance exercise. No interaction was observed (F = 1.01, *p* = 0.34, partial $\eta^{2}=0.09$). Post hoc analysis revealed that blood flow at 10 W (t = 2.37, *p* = 0.04) and 15 W (t = 3.27, *p* < 0.01) was significantly lower during continuous versus intermittent exercise.

When comparing the MAP at the two different power outputs during continuous and intermittent resistance exercise (Figure 2B), repeated-measures ANOVA indicated a main effect of power on the MAP (F = 8.70, *p* < 0.01, partial $\eta^{2}=0.49$) such that the pressure increased with power. Importantly, a main effect of exercise configuration was also observed (F = 24.94, *p* < 0.01, partial $\eta^{2}=0.74$). Post hoc analysis indicated that the MAP was significantly lower during intermittent exercise than during continuous exercise at 10 W (t = 2.75, *p* < 0.01) and 15 W (t = 3.69, *p* < 0.01). No interaction between the power and exercise configuration was observed (f = 1.16, *p* = 0.31).

When comparing leg vascular resistance at the two different power outputs during continuous and intermittent resistance exercise (Figure 2C), repeated-measures ANOVA indicated a main effect due to the configuration (F = 19.33, *p* < 0.01, partial $\eta^{2}=0.68$), but no main effect due to power (F = 0.44, *p* = 0.525, partial $\eta^{2}=0.05$). Post hoc analysis indicated that the resistance was significantly greater during continuous exercise than intermittent exercise at 10 W (t = 3.96, *p* < 0.01) and 15 W (t = 3.60, *p* = 0.04). No significant interaction between the power and configuration was observed (F = 0.14, *p* = 0.72, partial $\eta^{2}=0.02$).

The effect of interrepetition rest on maximum sustainable power (P_SUS_): Ten subjects performed a graded exercise test in which the subjects performed 3 min bouts of continuous and intermittent resistance exercise at increasingly heavier weights until failing to complete a 3 min stage (Figure 3). During the continuous resistance configuration, the subjects were able to lift 14.7 ± 0.7 kg at a contraction frequency of 30 contractions per minute for 3 min. During the intermittent configuration, the subjects were able to lift 40.0 ± 2.8 kg at a contraction frequency of 15 contractions per minute for 3 min (t = 3.04, *p* = 0.01, partial $\eta^{2}=0.96$). When considered in terms of power, a paired t-test indicated that the subjects were able to sustain a significantly higher power output during intermittent resistance exercise than during traditional continuous resistance exercise (29.5 ± 2.1 vs. 21.7 ± 1.2 W, respectively; t = 3.04, *p* = 0.01, partial $\eta^{2}=0.96$).

### 3.2. Protocol #2: Impact of Interrepetition Rest on the Blood Flow and Tolerance to Heavy-Weight Resistance Exercise

The subjects performed heavy (80% 1RM) resistance exercises until task failure with a continuous configuration (c80%) and an intermittent configuration (i80%). The subjects also performed a lighter continuous resistance exercise (40% 1RM) until task failure (c40%) to provide a power-matched comparison with the heavier, yet less-frequent, intermittent configuration.

Impact of interrepetition rest on exercise tolerance: As illustrated in Figure 4A, repeated-measures ANOVA indicated a significant effect of exercise configuration on the total repetitions that subjects could perform (F = 45.06, *p* < 0.01, partial $\eta^{2}=0.83$), with subjects performing significantly (t = 2.80, *p* = 0.02) more repetitions of i80% (26.5 ± 5.3 repetitions) than during c80% (16.8 ± 2.1 repetitions). As illustrated in Figure 5B, repeated-measures ANOVA indicated a significant main effect of exercise configuration on the total work that could be performed (F = 6.18, *p* < 0.01, partial $\eta^{2}=0.41$), with post hoc analysis indicating that subjects performed significantly more work during i80% (t = 2.8, *p* = 0.02) and c40% (t = 4.47, *p* < 0.01) trials than during the c80% trial. Notably, subjects completed the same total work during the i80% and c40% trials (t = 0.04, *p* = 0.97), which, by design, were performed at the same power output.

Impact of interrepetition rest on hemodynamics during heavy-weight resistance exercise: Given the differing amounts of time that the exercise configurations were performed, hemodynamic variables were binned and averaged according to the percent of total time (10% bins) in order to compare the hemodynamic responses across exercise configurations. As illustrated in Figure 5A, repeated-measures ANOVA of the blood flow response revealed significant main effects due to the configuration (F = 30.17, *p* < 0.01, partial $\eta^{2}=0.79$) and time (F = 37.25, *p* < 0.01, partial $\eta^{2}=0.82$), while no significant interaction was detected (F = 1.27, *p* = 0.22). Post hoc analysis indicated that the overall blood flow response during c80% was significantly lower than the responses observed during i80% (F = 51.59, *p* < 0.01) and c40% (F = 45.43, *p* < 0.01). Meanwhile, the overall blood flow response to i80% was significantly greater than the response to c40% (F = 0.7.67, *p* = 0.02), despite being performed at the same power output.

As illustrated in Figure 5B, repeated-measures ANOVA indicated a significant interaction between the exercise configuration and time (F = 5.57, *p* < 0.01), and significant main effects due to the configuration (F = 27.40, *p* < 0.001) and time (F = 10.42, *p* < 0.001) on the vascular resistance. Post hoc analysis indicated that vascular resistance was significantly lower in response to i80% and than c80% (F = 27.35, *p* < 0.01).

As illustrated in Figure 5C, repeated-measures ANOVA indicated a significant interaction between the exercise configuration and time (F = 6.94, *p* < 0.01, partial $\eta^{2}=0.46$) and significant main effects due to the configuration (F = 8.36, *p* < 0.01, partial $\eta^{2}=0.51$) and time (F = 4.20, *p* < 0.01, partial $\eta^{2}=0.34$) on the total concentration of heme found in the thigh using NIRS. Post hoc analysis indicated that the overall change in [Heme] in response to exercise was significantly greater in response to i80% (F = 14.44, *p* < 0.01) and c40% (F = 10.17, *p* < 0.01) than c80%, while i80% and c40% were not significantly different (F = 0.89, *p* = 0.37).

Finally, as illustrated in Figure 5D, repeated-measures ANOVA indicated a significant interaction between the exercise configuration and time (F = 6.84, *p* < 0.01, partial $\eta^{2}=0.46$), and a significant main effect due to the time (F = 18.14, *p* < 0.01, partial $\eta^{2}=0.69$) but not configuration (F = 1.25, *p* = 0.31, partial $\eta^{2}=0.14$) on the percent oxygenation of heme was found in the thigh using NIRS. As indicated by the significantly greater desaturation at task failure (*p* < 0.001), muscle heme desaturated more toward the end of c80% than during i80% or c40%.

## 4. Discussion

The present study investigated the impact of brief, interrepetition rest (2 s) on the blood flow response and exercise tolerance to knee-extension resistance exercise. Overall, the data indicated that 2 s of interrepetition rest was sufficient to substantially decrease vascular resistance, increase muscle blood flow and increase various aspects of exercise tolerance during resistance exercise. These findings and their implications are discussed in greater detail below.

### 4.1. How Does Interrepetition Rest Impact Hemodynamics during Resistance Exercise?

As mentioned, resistance exercise is associated with high intramuscular pressures that impinge upon the skeletal muscle vasculature, increasing vascular resistance and decreasing blood flow in the process [10,12]. Blood flow increases abruptly between contractions in a process known as “reactive hyperemia”, which is when intramuscular pressures decrease [18,19,24,25].

In protocol #1, we tested the hypothesis that interrepetition rest, with its accompanying reactive hyperemia, would result in less vascular resistance than traditional continuous resistance exercise, thereby facilitating greater flow at a given power output. As illustrated in Figure 2A, the 2 s of rest and reactive hyperemia between each contraction were sufficient to increase leg blood flow and decrease vascular resistance during intermittent resistance exercise compared with continuous resistance exercise at the same power output. This was consistent with results from Broxterman et al. [26], who found that increasing the duty cycle of a contraction (i.e., how long a contraction lasted) reduced blood flow, and subsequently, exercise tolerance at a given power output, supporting the notion that it is the contraction itself that is increasing resistance.

Several previous studies indicated a lower blood pressure response to resistance exercise with interrepetition rest than to the same exercise in a continuous configuration [5,6]. For example, Iglesias-Soler et al. [5] reported that a traditional continuous configuration of squats, with continuous repetitions performed in sets separated by 180 s of rest, elicited a 26% increase in systolic blood pressure, while the same exercise with the rest distributed between each repetition only elicited a ~11% increase in systolic blood pressure. As illustrated in Figure 2, we also observed a blunted blood pressure response with interrepetition rest, which was at least partially explained by lower leg vascular resistance. Therefore, we confirmed that resistance exercise with just 2 s of interrepetition rest elicited lower vascular resistance and greater leg blood flow than traditional continuous exercise.

### 4.2. Does Interrepetition Rest Increase the Power That Can Be Sustained during Resistance Exercise?

Since blood flow is necessary for the recovery of creatine phosphate and for controlling the concentration of fatigue-inducing metabolites [7,8], we next examined whether subjects could achieve a higher P_SUS_ (i.e., the highest power they could sustain for 3 min) with interrepetition rest than during traditional continuous contractions. As illustrated in Figure 3, the subjects were able to lift more weight and achieve a greater P_SUS_ with the intermittent configuration than with the traditional continuous configuration. Clearly, the intermittent configuration required half of the contraction frequency (i.e., greater contraction cycle duration) of the continuous protocol, but this reduction in contraction frequency was exceeded by the more than doubling in mass lifted with the intermittent configuration.

Results from previous studies support the beneficial impact of rest between repetitions on power output during resistance exercise [1,3,4]. Lawton et al. [1] found that allowing subjects to take 20 s rest between repetitions of bench press resulted in a smaller decrease in peak power over the course of a six-repetition set. Tufano et al. [4] reported similar findings with interrepetition rest of 12 s minimizing the decrease in power observed across 36 repetitions of back squats at 75% 1RM. Thus, rest between repetitions, whether it be 2 s or 20 s, appears to be sufficient to increase muscle blood flow and the sustainability of a given power output.

Based on our findings, it seems likely that augmented muscle blood flow has a role in the beneficial effect of interrepetition rest on muscle endurance (i.e., P_SUS_) during resistance exercise. Although not measured in this study, previous studies indicated that blood flow facilitates greater rates of creatine phosphate resynthesis and fatigue recovery [7,8,28]. For example, Hammer et al. [28] demonstrated a rapid increase in fatigue with cuff-induced arterial occlusion (i.e., reduced blood flow by inflating a blood pressure cuff) that was rapidly ameliorated when blood flow was restored by removing the occlusion. It seems possible that the high intramuscular pressures present during resistance exercise in the current study functioned like the occlusion cuff in the Hammer et al. study [28] such that removing the intramuscular pressures associated with the contraction by taking breaks between contractions permitted increased muscle blood flow and exercise tolerance. It is important to note that since we only measured bulk blood flow through the femoral artery, it is unclear how much of the blood flowing through the femoral artery during each contraction made it to the exercising muscle and how much was diverted down a path of lesser resistance to inactive tissue.

### 4.3. Does Interrepetition Rest Impact Tolerance to Heavy-Weight Resistance Exercise?

Having observed that interrepetition rest increased the power that could be sustained during resistance exercise (Figure 3), we next examined the impact of the same brief interrepetition rest on the number of repetitions a person could perform during a heavy resistance exercise that is typical of strength training [29]. Subjects performed as many heavy-weight (80% 1RM) repetitions of knee extensions as possible with a continuous configuration (c80%) and with an intermittent configuration (i80%). A third trial of 40% 1RM with no interrepetition rest (c40%) was also included to match the power output of the i80% trial. As illustrated in Figure 4, subjects performed significantly more repetitions of 80% 1RM and significantly more work when 2 s of rest were included between repetitions (i80% vs. c80%, *p* = 0.02). Subjects performed approximately the same total amount of work during the c40% and i80% trials (*p* = 0.97), which, despite being different weights and frequencies, had the same overall power output. As will be described below, we believe that this increased exercise tolerance was related to improved muscle blood flow during the intermittent configuration.

### 4.4. Does Interrepetition Rest Impact Muscle Blood Flow during Heavy-Weight Resistance Exercise?

Salvador et al. [30] recently reported that a 30 s rest in the middle of a set of resistance exercises resulted in significantly lower concentrations of lactate within the blood immediately after exercise. Others, such as Byrd et al. [1,31], speculated that similar reductions in blood lactate with interrepetition rest were associated with improved muscle blood flow and oxygen delivery. To our knowledge, ours is the first study to directly test the impact of interrepetition rest on muscle blood flow and oxygen delivery.

As illustrated in Figure 5, we measured leg blood flow, vascular resistance and quadriceps muscle oxygenation during the heavy-weight resistance exercise trials. In agreement with the hypotheses of previous researchers [31], leg blood flow increased significantly more during heavy-weight (80% 1RM) resistance exercise when 2 s of interrepetition rest was allowed. While leg blood flow increased during all configurations, it is important to remember that an increase in bulk blood flow through the femoral artery does not necessarily mean that the blood reached the exercising muscle that needed it [32]. Indeed, despite the increase in leg blood flow during the c80% trial, the [Heme] measured using near-infrared spectroscopy (NIRS) in the vastus lateralis significantly decreased (*p* < 0.05). Meanwhile, [Heme] significantly increased and remained significantly more saturated with oxygen during the i80% trial (*p* < 0.05).

It seems possible that intramuscular pressures remained sufficiently high during the c80% trial to impinge upon the vessels feeding the active muscle throughout the entirety of the exercise (i.e., the concentric and eccentric phases), resulting in blood being redirected toward inactive areas of the leg with less vascular resistance. In other words, while blood flow to the leg increased during the i80% trial, it did not result in increased blood flow present in the exercising muscle, likely because it was redirected down a path of lower resistance to the inactive muscles. Meanwhile, interrepetition rest allowed for the flow through the femoral artery to actually reach the exercising muscle, augmenting [Heme], oxygen delivery and exercise tolerance in the process.

### 4.5. Future Directions

Resistance exercise is commonly associated with very large increases in blood pressure, which may be problematic for individuals with angina or already high blood pressure [20]. As illustrated in Figure 2B, a brief rest between repetitions was associated with a lower MAP during exercise. Paulo et al. [6,20] recently measured the blood pressure response to resistance exercise with varying durations of rest between repetitions in healthy and hypertensive adults. In agreement with our findings, Paulo et al. reported a smaller increase in blood pressure and strain on the heart when 22 s of rest were included after every 5 repetitions of a 15-repetition set. Thus, interrepetition rest may be a simple and effective way to reduce the hypertensive risks posed by traditional resistance exercise. More research regarding the effectiveness of interrepetition rest or cluster set protocols on the blood pressure responses of different populations is warranted.

The use of interrepetition rest may also be beneficial for strength training programs by increasing the relative intensity of repetitions (i.e., %1RM) that can be repeatedly performed or by increasing the total volume of a workout. However, with reductions in blood flow (i.e., blood flow restriction training) significantly augmenting the hypertrophic response to light exercise [33,34], it is unclear whether increasing flow with interrepetition rest will elicit the same hypertrophic responses as traditional, continuous resistance exercise. Two studies examining the impact of cluster set training on muscle hypertrophy and protein synthesis provide insight in this regard. Salvador et al. [30] reported that allowing subjects to take a single 30 s rest in the middle of a 10-repetition set did not significantly blunt the protein synthesis rate elicited by resistance exercise. Additionally, Oliver et al. [35] reported that individuals who performed 12 weeks of resistance training with 60 s of rest in the middle of 10 repetition sets exhibited the same changes in muscle fiber type and greater increases in power and strength compared with those who performed an equal amount of continuous resistance exercise. While these cluster set data suggest that performing an equal amount of resistance exercise with interrepetition rest might elicit the same, if not greater, adaptations as traditional continuous exercise, the impact of the increased number of repetitions associated with interrepetition rest has yet to be examined. Studies examining the impact of work-matched (same number of repetitions) and effort-matched (e.g., maximum repetitions to exhaustion) resistance exercise with or without interrepetition rest on neuromuscular and vascular adaptations, as well as injury rates, will help to determine the utility of resistance training with interrepetition rest.

### 4.6. Experimental Considerations

Previous studies demonstrated greater reactive hyperemia among physically active individuals than sedentary individuals [24,25]. With a sample size of 10 subjects from a variety of training backgrounds, it is unknown how well these results apply to the broader population or specific patient populations. Therefore, caution should be taken when extrapolating these results to other populations. Future studies should investigate whether the benefits of interrepetition rest vary between individuals of various training and health backgrounds.

The order of continuous and intermittent bouts at a given power output was not randomized in the first protocol due to concerns about the accumulation of fatigue with repeated bouts. With pilot studies indicating that task failure would occur at a lighter weight during continuous exercise than intermittent exercise, the continuous bouts of exercise were always performed before the intermittent bouts, thereby biasing the potential impact of fatigue induced by previous bouts toward the intermittent exercise (i.e., more work was performed prior to an intermittent bout of exercise compared with a bout continuous exercise of equal power). It is possible that the impact of interrepetition rest on P_SUS_ may have been blunted by this lack of randomization. Additionally, while at least two days of rest separated each visit, it is possible that fatigue from previous visits was present during data collection. Nevertheless, with all comparisons being made within subjects and within days, it seems unlikely that pre-existing fatigue would account for the differences we observed between continuous and intermittent exercise.

The typical stretch-shortening cycle present during continuous resistance exercise was likely missing during the intermittent protocol. This may have caused the loss of contractile muscle abilities, which may have led to the loss of power and more significant energy expenditure, which could have contributed to the augmented blood flow response during intermittent exercise.

With the leg attached to a cable that not only moved forward and backward, but also side-to-side, it is possible that extraneous movements added to the metabolic cost and blood flow response to exercise and varied from person to person. Additionally, by setting the displacement distance of the weight to 15 cm, subjects with differing leg lengths extended their legs to varying degrees, which may have affected how the contraction was performed. Nevertheless, all comparisons in this study were made within subjects, making the variability in extraneous movements and magnitude of knee extension of lesser importance. Likewise, differences in lower leg mass could affect the actual amount of work performed by each subject through a contraction cycle. Nevertheless, with all comparisons being within subjects, the effect of lower leg mass on work was constant for each configuration and was unlikely to affect the main conclusions of the study.

## 5. Conclusions

Brief rest between repetitions of resistance exercise effectively decreased vascular resistance, increased muscle blood flow and increased exercise tolerance. Interrepetition rest may be a useful intervention to decrease the hemodynamic burden and improve tolerance to resistance exercise.

## Figures and Tables

**Figure 1 medicina-58-00822-f001:**
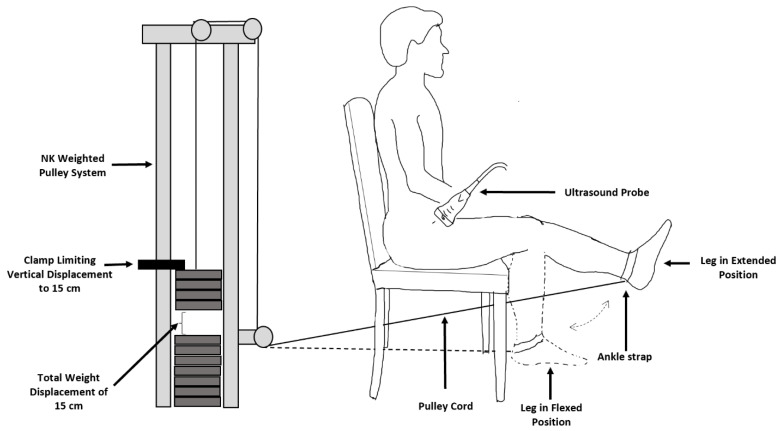
Illustration of Experimental Setup for Knee Extension Exercise. Subjects performed concentric and eccentric knee extension exercises with their ankle attached to a cable-based pulley system. The vertical displacement of the pulley was limited to 15 cm by placing a clamp 15 cm above the weight stack. The pulley system was adjusted so that knee flexion was 90° at rest. Leg blood flow in the common femoral artery was measured during knee extension with a Doppler ultrasound.

**Figure 2 medicina-58-00822-f002:**
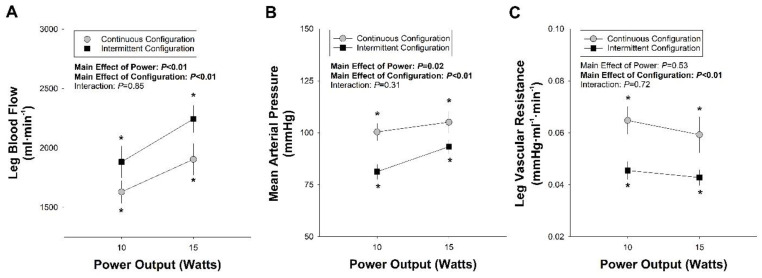
Impact of Interrepetition Rest on Hemodynamics during Single-Leg Knee Extension Resistance Exercise. (**A**) Leg blood flow, (**B**) mean arterial pressure and (**C**) leg vascular resistance during single-leg knee extension with (i.e., intermittent configuration) or without (i.e., continuous configuration) interrepetition rests. * Significantly different (*p* < 0.05) from the same power output under another configuration.

**Figure 3 medicina-58-00822-f003:**
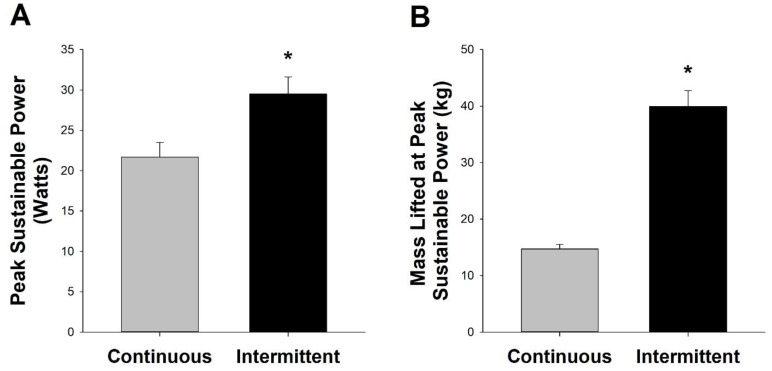
Impact of Interrepetition Rest on Peak Sustainable Power (P_SUS_) during Single-Leg Knee Extension Resistance Exercise. (**A**) P_SUS_ during 3 min of continuous (i.e., no rest between repetitions) and intermittent (i.e., 2 s rest between repetitions) knee extension exercise. (**B**) Mass lifted during trials eliciting P_SUS_. * Significantly different (*p* < 0.05) from the continuous trial.

**Figure 4 medicina-58-00822-f004:**
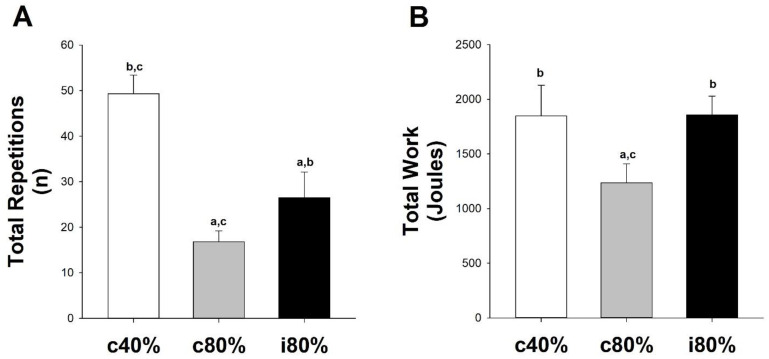
Impact of Interrepetition Rest on Exercise Tolerance to Heavy Weight Single-Leg Knee Extension Resistance Exercise. (**A**) Total repetitions performed and (**B**) total work during knee extension performed continuously at 40% (c40%) and 80% (c80%) of one-repetition max (1RM), as well as intermittently with 2 s of interrepetition rest at 80% (i80%) 1RM. a: significantly different than the c40% trial; b: significantly different than the c80% trial; c: significantly different than the i80% trial.

**Figure 5 medicina-58-00822-f005:**
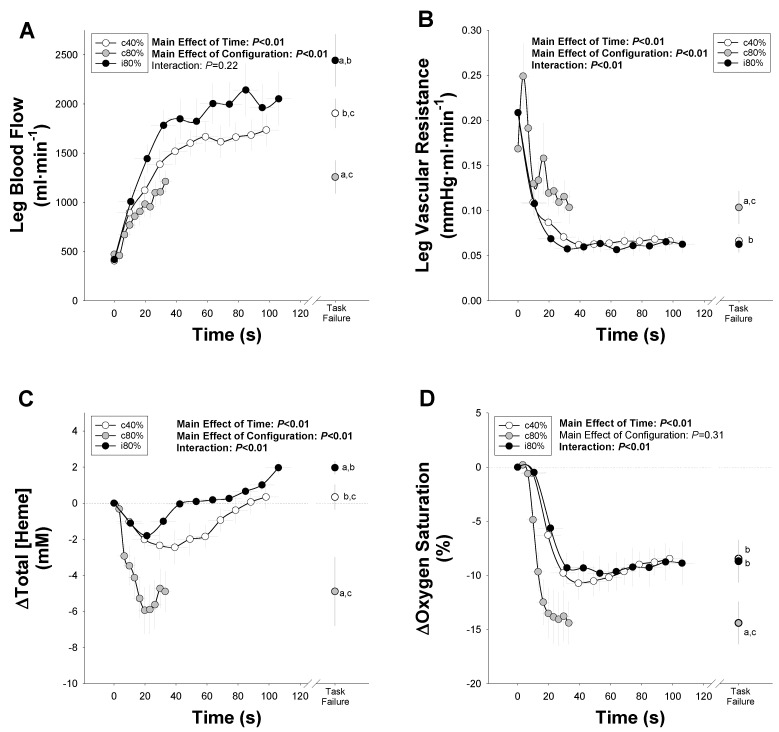
Impact of Interrepetition Rest on Hemodynamics during Heavy Single-Leg Knee Extension Resistance Exercise Performed to Task Failure. (**A**) Leg blood flow through the femoral artery, (**B**) leg vascular resistance, (**C**) the change from rest in the total concentration of heme ([Heme]) of the exercising thigh and (**D**) the change in oxygen saturation of heme in the exercising thigh during knee extension performed continuously at 40% (c40%) and 80% (c80%) of one-repetition max (1RM), as well as intermittently with 2 s of interrepetition rest at 80% (i80%) 1RM. a: significantly different than the c40% trial; b: significantly different than the c80% trial; c: significantly different than the i80% trial.

## Data Availability

Data are available upon request to the corresponding author.

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
