# Peer review of "Impact of Interrepetition Rest on Muscle Blood Flow and Exercise Tolerance during Resistance Exercise"

_medicina, 2022, doi:10.3390/medicina58060822_

Round 1

Reviewer 1 Report

Dear Authors,

In the manuscript submitted for review, authors investigate the impact of brief bouts of rest (2-s) between repetitions of resistance exercise on muscle blood flow and exercise tolerance. The value of the work is that the improved exercise tolerance and cardiovascular strain, associated with interrepetitions rest, could be related to decreases in vascular resistance and improved blood flow to the exercising muscle is directly tested.

The only concern is that only 10 subjects were recruited for the study, and this unfortunately impairs the soundness of your results.

Lines 96-99: you state that “Subjects from a variety of self-reported training backgrounds (e.g., sedentary, endurance trained, or strength trained) were used for this study to help generalize the results to a broader population”. However, with 10 subjects only, results could not be any way generalised, you should confirm your statement with some appropriate citations.

Minor comments:

Line 74: it should be clarified that each protocol consists of two exercise mode, continuous and intermittent, other ways the explanation at Lines 81 84 is difficult to understand.

Reviewer 2 Report

The present study aims to examine the effect of including interrepetition rest during resistance exercise on muscle blood flow and exercise tolerance. The introduction correctly justifies the study, which seems pertinent in view of the current gap. Methods are clearly and correctly described, and the results are well detailed. However, several aspects need to be clarified, which are detailed below.

INTRODUCTION:

P1 Ln30. It would be recommended for the authors to include a brief description with the basic characteristics of the resistance exercise to which they refer, in terms of percentage of loads or number of repetitions, as well as its negative effects on muscular blood flow.

P1 Ln37-39 and P2 Ln68-71. The authors mention twice the hypothesis of the study. Would it be possible to mention it only at the end of the introduction, next to the objective of the study?

METHODS:

P2 Ln72. Please, detail which specialist was in charge of supervising the exercises and carrying out the assessments. Likewise, describe the setting where the study was conducted.

P2 Ln72. For a better structure of the article, it is recommended to include a subheading where the study variables and data collection methods are listed, since the variables are cited in P2L77-81, and the data collection methods appear later.

P2 Ln83. The authors establish a rest period between protocols of 2 days. Could you provide any reference to previous studies that support it?

P2 Ln89. Is the study previously registered in any clinical trials database? This aspect should be clarified for the possible study of publication bias.

P2 Ln93. Why was an adequate sample size calculation not carried out in order to achieve reliable statistical power? Ten subjects seem more like a pilot study. In this way, it would be necessary to discuss at the end of the study that the results obtained in it must be handled with caution.

P5 Ln177-182. Regarding the influence of fatigue on the results of protocol #1, would it not have been more recommended to randomise the order in which the tests were performed (continuous and intermittent) as in protocol #2? They should be discussed later.

P6 Ln212-217. Please provide reference for this paragraph.

P6 Ln 224. Please provide reference for this calculation of vascular resistance.

P6 Ln 225-233. Please provide reference for this paragraph.

P6 Ln 238-245. Provide references for these calculations, as well as the units in which they are expressed.

DISCUSSION

P9 Ln360. Include “and exercise tolerance”?

CONCLUSION

P14 Ln 551-552. These two lines do not correspond to any conclusion of the study and should be removed.

FIGURES

Figure 2 is resolution low.

REFERENCES

Ten references are more than 30 years old. Please update with recent studies unless these references are essential.

SPECIFIC COMMENTS

P1 Ln23-26. Modify the font format.

P2 Ln46; P6 Ln239 and Ln 247. The beginning of the paragraph has extra space.

P10 Ln397. They instead of The?

P12 Ln466. Reported instead of eported?

Author Response

This manuscript is a resubmission of an earlier submission. The following is a list of the peer review reports and author responses from that submission.

Round 1

Reviewer 1 Report

In the manuscript submitted for review, authors investigate the impact of brief bouts of 10 rest (2-s) between repetitions of resistance exercise on muscle blood flow and exercise tolerance. This is an interesting object of study and may shed more light on the relationship between rest bout duration workload. However, I believe that different aspects of the paper should be corrected and/or clarified.

In particular protocol two does not seem to add value to your work and is not pertinent to the aim of the study. You should consider deleting it unless strong evidence of its importance and relation with the aim of the work are brought about. Most importantly, the rationale for the comparison of intermittent with continuous and of resistance with endurance exercise has not been explained, the reason they add value to your work needs to be supported by convincing explanations.

Moreover, it is stated that subject number is 16, while they were divided in two groups (10+6) performing different exercise protocols, therefore protocol 1 and 3 have been carried out by 10 subjects only and this obviously impairs the soundness of your results.

In my opinion the optimal solution to improve the quality of your work would be to delete protocol 2 and to carry out protocol 1 and 3 with more subjects.

Here is the list my point-by-point comments.

Abstract, Line 12: specify number of subjects.

Abstract, Line 23: specify what “leg blood flow muscle [Heme]” is.

Line 54: explain what “critical torque” is.

Line 62: to be more specific replace "following" with "after the cessation of".

Lines 98-99: subjects’ number is not 16, they are two groups of 10 + 6 performing different test protocols, this should be put forward and their description needs to be presented separately.

Lines 186-187; 322-357: the rationale for the comparison of resistance and endurance exercise has not been explained, you should explain why it adds value to your work and support its importance.

Lines 268-269: protocol 2 has been carried out by 6 subjects only, in this case nonparametric statistics analysis is more appropriate.

Lines 287-289, 298-302, 384-357, 480-486, 502-509: summarise all figures captions, they should be 2-3 lines long and do not contain explanation of the protocol.

Lines 291-292: the advantages of intermittent vs continuous endurance exercise are well known since long time (doi: 10.1055/s-2000-3782), you need to highlight what innovative information your results bring.

lines 553-554: there is no need to repeat the same concept twice.

Thank you for giving me the opportunity of reading this interesting manuscript.

Author Response

Thank you for your careful review of our manuscript.  We have carefully considered your suggestions and feel that the manuscript is much better and clearer for it.  Thank you. 

Based upon feedback from both reviewers, we removed the previous Protocol #2, which compared blood flow during resistance exercise to dynamic exercise.  While we initially thought this was novel and interesting, we agree with the reviewers that it was superfluous and distracting. We now only present 2 protocols with Protocol #1 investigating the impact of interrepetition rest on blood flow and exercise tolerance during high-repetition, low-weight resistance exercise and Protocol #2 investigating the impact of interrepetition rest on blood flow and exercise tolerance during heavy-weight, low-repetition resistance exercise.  Importantly, each protocol was performed on the same 10 subjects.  Please note that the exclusion of previously included data has resulted in a re-assignment of Figure numbers. 

Again, we are very grateful for your careful review and think that this paper is much better because of your insight.  We address each reviewer’s specific concerns below.  Thanks.

Reviewer #1

  • Endurance Exercise Protocol Removed
    • As mentioned above, we have followed your suggestion and removed the protocol looking at endurance exercise. While interesting, it was tangential to the research topics.  Removing this protocol simplified the study design and write up.  Thank you for your suggestion.
  • Line 98-99
    • Subject numbers.
      • Due to throwing out the original protocol #2, we now have the same 10 subjects in each reported protocol. We now indicate this in the abstract and methods. With 10 subjects, we are able to detect meaningful differences in blood flow, and exercise tolerance in each protocol.
    • Line 23:
      • Instead of describing muscle [heme] in the abstract, we have removed it from the abstract. It is described in greater detail in the body of the manuscript.
    • Line 54
      • Critical torque is the maximal steady state threshold. We now indicate this in the introduction.
    • Novelty of experiment
      • Most studies with cluster sets or interrepetition rest hypothesize that improved muscle blood flow is the reason for improved performance, but it has never been measured until this study. As far as we know, this is the first study to measure muscle blood flow and oxygenation during resistance exercise with and without interrepetition rest.  While the findings are somewhat predictable, this is the first bit of empirical evidence to directly examine blood flow during this type of exercise and is therefore necessary and novel.  We now clarify this in the introduction and discussion.
    • Figure Legends
      • We have decreased the length of the figure legends. However, with four panels in Figure 4, the length of this legend is, by necessity, longer than the rest. Thanks.

Thank you for your careful and constructive reviews of our manuscript.  You have challenged us and helped us to present a clearer manuscript.  We appreciate your efforts to help us. 

Reviewer 2 Report

This study examines the impact of interrepetition rest on muscle blood flow and exercises tolerance during resistance exercise. Studies with obvious practical significance are often needed, and they are quite trendy. From that point of view, this study is intriguing and relevant. However, several significant flaws are noticed in the study rationale, methods applied, and results. Furthermore, the Discussion is inadequately written. More detailed comments are presented further.

Introduction: 

  • The Introduction is relatively well written; however, there is an evident lack of studies previously examining intermittent repetitions. There are numerous studies investigating cluster sets. The authors used too much Introduction explaining the physiological aspect of intermittent repetitions rather than focusing on the previous studies demonstrating this phenomenon. 
  • Based on the previous comment, the rationale for this study is not clear and should be backed up with the previous studies examining this.

Methods: 

Subjects:

  • Can the authors justify why the subjects ranged from sedentary to highly trained? 
  • How did the authors test the level of physical activity to declare someone is sedentary or trained?
  • Can authors elaborate on splitting the sample into three protocols 10 + 6 + 10 participants?

Familiarization:

  • Can authors elaborate or cite a study that performed extension from 90 degrees to 15cm? 
  • For this kind of testing, the use of isokinetic dynamometer was necessary. There are a lot of degrees of freedom when using the pulley device in regards to the dynamometer. This is particularly important since a lot of participants were not strength trained. Furthermore, how did the authors limit the movement to 15cm?
  • A picture or a graph of the single-leg knee extension on this pulley device would greatly benefit this manuscript.

Protocol #2:

  • Can authors further explain “the six subjects also reported to the lab on a separate day to perform…“ This was not previously reported in the Experimental approach to the problem. “during lighter-weight, higher-repetition resistance exercise.” Endurance was not mentioned before.

Calculation of Work and Power:

  • Did the authors include the mass of the lower leg in the calculus? 
  • I’m not convinced that this is a proper calculation of the power output. Namely, participants perform an angular motion of the lower leg when performing a 15cm knee extension (from the 90 degrees angle). The presented calculus was, however, for the linear motion of the weights. This is a highly unprecise method since the pulley cable can be moved in all directions. This is particularly detected in untrained subjects.
  • This brings me back to the previously raised issue. This exercise should be performed on the isokinetic dynamometer. In this particular case, 15cm extension for someone who is 195cm tall (with the lower knee of 30cm) and 155cm tall (with the lower knee of 20cm) is a vast difference in the mechanical work performed. Also, the movement pattern is different. When the knee extends to more degrees - the lower leg is lifted to the higher position, which influences the pulling device (e.g., increasing the friction that was not calculated) and participant as well.

Statistical analysis:

  • Since there are only 6 participants in some tests and 10 in others, the use of parametric tests is questionable? Furthermore, the normality of the data was not tested. Authors should consider using non-parametric statistical tests or present good evidence for using parametric tests.
  • As I understand, protocols #2 and #3 were performed on different participants. I’m not sure how Repeated-measures ANOVA was used?

Results: 

  • The authors only presented p values in the analysis. Please indicate T or F values for the t-tests and ANOVA-s (respectfully) and some effect size results (e.g., eta squared).

Discussion: 

Does Interrepetition Rest Increase the Power that Can Be Sustained during Resistance Exercise?:

  • Authors should elaborate their findings in more detail, not just repeating the results and mentioning that others got the same results. 

How Does Interrepetition Rest Impact Hemodynamics during Resistance Exercise?:

  • Similar to previous, many results repeat and not much of a discussion.

Does Interrepetition Rest Impact Muscle Blood Flow during Heavy-Weight Resistance Exercise?:

  • Same as previous…24 lines of text and only one citation  

How Might Muscle Blood Flow Influence Tolerance to Resistance Exercise?

  • Similar as previous. Authors usually mention a study, then explain the results of this study, and finally repeat the current study results.

Practical Applications:

  • Practical Applications should be concise with straightforward suggestions for the coaches/sports practitioners. There is too much Discussion within this chapter. Some of that should be used in the previous chapters. 
  • In intermittent exercises, the stretch-shortening cycle is missing, contributing to the loss of some contractile muscle abilities. Consequently, this can lead to the loss of power and more significant energy expenditure. This is particularly important to prevent in highly trained athletes, and it should be noted within this study.

Limitations of the study should be presented as a chapter:

Figures

  • Figure 2 resolution is low
  • Figure 5 resolution is low

Specific comments:

Line 43, 57…The beginning of the paragraph has extra space.

Line 119…“Determination of Knee Extension 1-Repetition Maximum (1RM)” the heading? It seems to be out of the place. 

Author Response

Thank you for your careful review of our manuscript.  We have carefully considered your suggestions and feel that the manuscript is much better and clearer for it.  Thank you. 

Based upon feedback from both reviewers, we removed the previous Protocol #2, which compared blood flow during resistance exercise to dynamic exercise.  While we initially thought this was novel and interesting, we agree with the reviewers that it was superfluous and distracting. We now only present 2 protocols with Protocol #1 investigating the impact of interrepetition rest on blood flow and exercise tolerance during high-repetition, low-weight resistance exercise and Protocol #2 investigating the impact of interrepetition rest on blood flow and exercise tolerance during heavy-weight, low-repetition resistance exercise.  Importantly, each protocol was performed on the same 10 subjects.  Please note that the exclusion of previously included data has resulted in a re-assignment of Figure numbers. 

Again, we are very grateful for your careful review and think that this paper is much better because of your insight.  We address each reviewer’s specific concerns below.  Thanks.

Reviewer #2:

  • Introduction
    • This paper is focused on the role of muscle blood flow in the improved endurance associated with interrepetition rest. It is intended to provide a cardiovascular perspective on a known phenomenon. Consequently, the introduction contains a lot of background in physiology than a paper focused only on functional outcomes.
    • While many have assumed that improved blood flow is the cause of improved performance with cluster sets, to our knowledge, this is the first paper to actually measure the blood flow response to resistance exercise with interrepetition rest, which makes it very novel.
    • Nevertheless, ww have added more references to the introduction regarding cluster sets. There is a substantial amount of literature supporting the benefit of cluster sets and interrepetition rest, albeit much longer rest than used in our study. We now refer readers to an excellent review article by Tufano et al 2016, describing the impact of cluster sets on power and fatigue.
  • Methods
    • Subjects
      • We did not want to limit our findings to only trained or untrained subjects. Consequently, subjects with a variety of training backgrounds were used in this study. As described in the methods, all subjects were familiarized with the exercise before data collection.  If working by the hypothesized methods, interrepetition rest should improve blood flow and performance in most healthy adults, not just trained or untrained adults.
      • There was understandable confusion regarding how many subjects participated in each protocol. By throwing out the previous Protocol #2, we now just include the same 10 subjects in both protocols.
    • Familiarization
      • The range of motion of the weight stack was limited to 15 cm by placing a clamp 15 cm above the weight stack, thereby limiting its vertical displacement. This was do to allow for the calculation of total work and to prevent the subjects from locking out their knees, which would temporarily alleviate the intramuscular forces.  We now address this in the methods section.
      • Clearly, a isokinetic dynamometer would have controlled the extraneous movements, which may have been greater in untrained subjects, better than our set up. Nevertheless, all of our comparisons are within subjects, and subjects likely exhibited the same degree of extraneous movements during each configuration. We now address this issue in the “Experimental Considerations” section.
    • Protocol #2:
      • This protocol has been removed from the study.
    • Calculation of work and power
      • The calculation for work and power we used was based upon the vertical movement of the weight stack. We did not include the mass of the lower leg or variability in friction throughout the range of motion.  These factors could affect the actual amount of work and power performed by each subject.  Nevertheless, all comparisons are made within subjects, such that each subject is their own control so that lower leg mass is constant for each subject. We now address this in the “Experimental Considerations” section.
    • Statistical Analysis
      • We now have 10 subjects in each protocol.
      • Repeated measures were performed because subjects performed multiple trials under different configurations.
      • We now report the T and F values in the results.
    • Discussion
      • We have increased the discussion of relevant physiology and studies in our discussion.
    • Practical Applications
      • We have changed this section to be identified as “Future Directions”, which more appropriately captures the intention of the paragraph.
    • Experimental Considerations
      • We have added a section entitled “Experimental Considerations” to address limitations and delimitations of the study.
    • Figures
      • We have provided a new copy of the figures
    • Line 119-Determination of Knee Extension…
      • We have re-ordered the contents of this section to first describe the familiarization and then the determination of 1RM, as opposed to the other way around.

Thanks for all of your helpful comments.  You have challenged us and helped us produce a better paper in the process.

Round 2

Reviewer 1 Report

Dear Authors,

I congratulate you on responding to all my comments, in the present form the work is well structured and presented.

Reviewer 2 Report

/